# A genetically encoded fluorescent biosensor for detecting itaconate with subcellular resolution in living macrophages

Pengkai Sun [1,2,3], Zhenxing Zhang[1,3], Bin Wang[1,2], Caiyun Liu[1,2], Chao Chen[1], Ping Liu[1] & Xinjian Li [1,2] ✉

Itaconate is a newly discovered endogenous metabolite promoting an anti-inflammatory program during innate immune response, but the precise mechanisms underlying its effect remains poorly understood owing primarily to the limitations of available itaconate-monitoring techniques. Here, we develop and validate a genetically encoded fluorescent itaconate biosensor, BioITA, for directly monitoring itaconate dynamics in subcellular compartments of living macrophages. Utilizing BioITA, we monitor the itaconate dynamics in response to lipopolysaccharide (LPS) stimulation in the context of modulating itaconate transportation and metabolism. Moreover, we show that STING activation induces itaconate production, and injection of AAVs expressing cytosolic BioITA into mice allows directly reporting elevation of itaconate level in activated macrophages derived from LPS-injected mice. Thus, BioITA enables subcellular resolution imaging of itaconate in living macrophages.

Itaconate is the most abundant metabolite produced by macrophages during inflammatory activation[1–3]. The mitochondrial enzyme aconitate decarboxylase 1 (ACOD1, also known as IRG1) highly expressed in lipopolysaccharide (LPS)-stimulated macrophages, generating massive itaconate from *cis*-aconitate[4] and the mitochondrial itaconate is exported to cytosol via the carriers that transport dicarboxylate and citrate[5]. Many factors, including LPS, type I and type II interferons, agonists of Toll-like receptor (TLR), and activation of transcription factor EB (TFEB) are able to induce IRG1 expression followed by itaconate biosynthesis in macrophages[4,6–10]. In activated macrophages, itaconate is converted to itaconyl-CoA by succinyl coenzyme A synthetase (SCS) and citramalyl-CoA hydrated from itaconyl-CoA is cleaved into pyruvate and acetyl-CoA by citramalyl-CoA lyase (CLYBL)[11,12].

Owing to the inclusion of an electrophilic α, β-unsaturated carboxylic acid, itaconate is able to modify protein cysteine residues by a Michael addition to form a 2,3-dicarboxypropyl adduct, a post-translational modification known as alkylation[10]. Recently, our study demonstrated that itaconate-mediated alkylation of TFEB, a key

transcription factor controlling lysosomal biogenesis, promotes the antibacterial ability of macrophages[13]. It has been reported that itaconate alkylated kelch-like ECH-associated protein 1 (KEAP1)[5], a central player in antioxidant response, JAK1 kinase[14], as well as NOD-, LRR-, and pyrin-domain-containing protein 3 (NLRP3) inflammasome[15], to activate anti-inflammatory programs in macrophages. Itaconate also impaired aerobic glycolysis by alkylating the glycolytic enzymes, including glyceraldehyde 3-phosphate dehydrogenase (GAPDH) and aldolase A (ALDOA)[16,17]. In addition, itaconate may function as an electrophilic stress regulator to induce ATF3-dependent stress responses[18], an inhibitor of TET DNA dioxygenases[19] and succinate dehydrogenase (SDH)[20] to limit inflammation.

A protein-based genetically encoded fluorescent sensor usually consists of a fluorescent protein and a sensing domain[21]. When a target molecule binds to the sensing domain, it changes the conformation of the sensor, resulting in the change of the sensor's fluorescent properties, for example brightness and fluorescence lifetime at a particular excitation/emission wavelength pair. Basing on these coupling events,

[1]CAS Key Laboratory of Infection and Immunity, CAS Center for Excellence in Biomacromolecules, Institute of Biophysics, Chinese Academy of Sciences, Beijing 100101, China. [2]College of Life Sciences, University of Chinese Academy of Sciences, Beijing 100049, China. [3]These authors contributed equally: Pengkai Sun, Zhenxing Zhang. ✉e-mail: lixinjian@ibp.ac.cn

changes of metabolite concentration can be monitored by detecting the changes of the sensor's fluorescent properties using a fluorescence microscope or a flow cytometer. Notably, the fluorescent sensors, when expressed with fusion of different subcellular localization sequences, are capable of measuring the rapid changes of free metabolite levels in different subcellular compartments, such as the mitochondria, nucleus, ER or lysosome. However, an itaconate fluorescent biosensor, by which monitoring the intracellular itaconate levels during inflammatory activation, remains to be developed.

Here, we report the development of a highly responsive, genetically encoded itaconate fluorescent biosensor, referred to as BioITA (biosensor for itaconate). We construct BioITA by coupling a circularly permuted green fluorescent protein (cpGFP)[22] into an itaconate-binding domain (IBD) derived from ItcR, a bacterial LysR-type transcriptional regulator (LTTR) family protein[23,24], followed by further optimizations. BioITA allows precise and convenient real-time detection of itaconate dynamics in subcellular compartments of activated macrophages, thereby providing a powerful tool for itaconate detection and bioimaging.

## Results

### Constructing and optimizing itaconate biosensors

The bacterial protein ItcR is unique in its ability to bind to itaconate[23]. Structurally, ItcR is a transcriptional regulator protein with an N-terminal DNA-binding helix-turn-helix motif and a C-terminal itaconate-binding domain (IBD)[24] (Supplementary Fig. 1a). The IBD was expressed in *Escherichia coli* and purified by affinity chromatography and gel filtration (Supplementary Fig. 1b, c). Multiangle light scattering (MALS) analysis showed that IBD appears as homodimer in solution (Supplementary

Fig. 1d). Furthermore, the crystal structure of IBD without or with itaconate binding was determined by multiple-wavelength anomalous diffraction and refined at a resolution of 3.25 and 1.48 Å, respectively (Fig. 1a and Supplementary Table 1). In line with the result of MALS (Supplementary Fig. 1d), the IBD proteins form a homodimer in both of the apo and itaconate bound states (Fig. 1a and Supplementary Table 1). Notably, alignment between the structures of IBD in the apo and itaconate bound states demonstrated that monomer in each dimer exhibited translocation of over 7 and 10 Å in the N- and C-termini, respectively (Fig. 1b). Because cpGFP is highly sensitive to conformational changes in the metabolite binding domain of the sensor, we hypothesized that chimera proteins with fusion of cpGFP to the ItcR IBD would provide a platform for the development of cpGFP-based itaconate biosensors.

To construct an itaconate prototype sensor, we fused a human centromere protein B (CENP-B) dimerization domain[25] at the C-terminus of cpGFP and connected this cpGFP-CENP-B protein to the IBD C-terminus (Supplementary Fig. 1e), which exhibits over 10 Å translocation upon itaconate binding (Fig. 1b). However, we only observed a modest fluorescence increase (~20%) from the prototype sensor upon itaconate addition (Supplementary Fig. 1f), suggesting that optimizations are required to enhance the performance of this prototype sensor.

Previous studies of metabolite biosensors imply that the linkers between metabolite binding domain and cpGFP are critical for the sensor's performance[26–29]. To improve the sensor response to itaconate, we expressed and purified a set of chimeras with different linkers of 1–8 amino acid residues connecting IBD to cpGFP and cpGFP to CENP-B (Supplementary Fig. 1g), and assayed for their fluorescence response to itaconate. Among them, we identified one chimera exhibited the most dramatic increase in fluorescence excited at

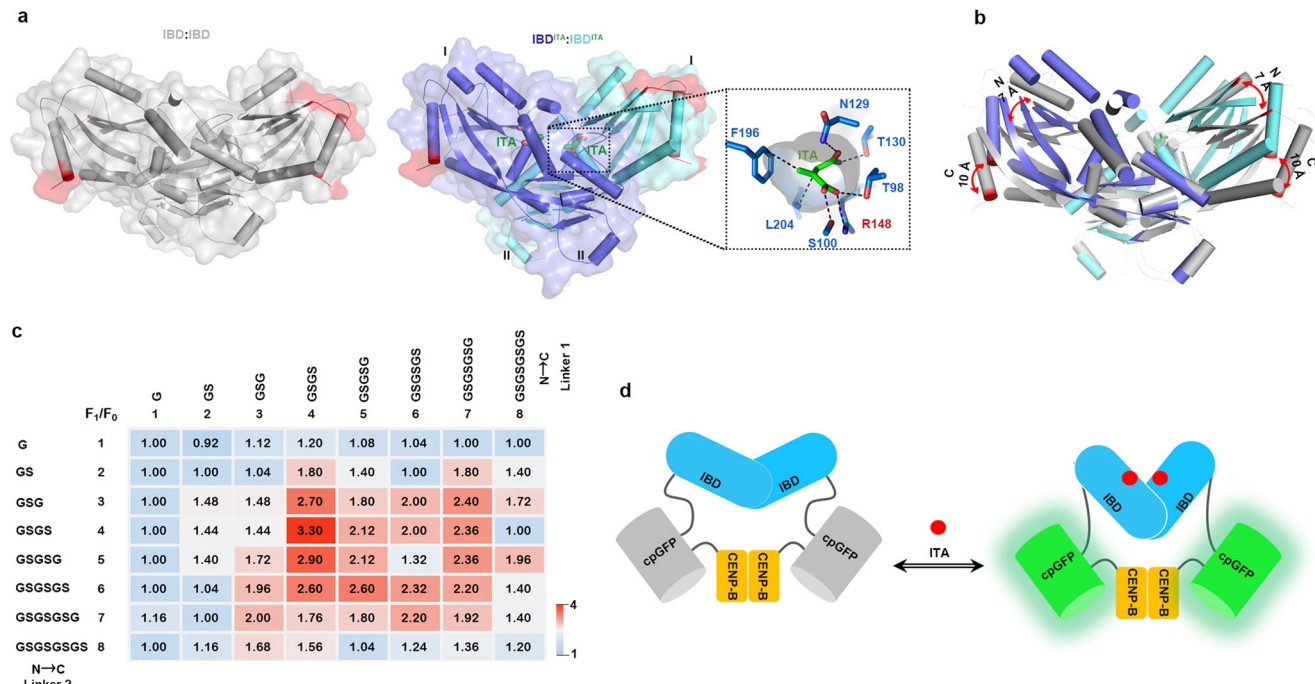

**Fig. 1 | Constructing and optimizing itaconate biosensor. a** Overall structure of IBD in apo (left, PDB: 7W08) or itaconate-bound (right, PDB: 7W07) state. The dotted lines circumscribed the zone of itaconate binding pocket and the key amino acid residues that interact with itaconate were shown. **b** Overlay and structural alignment of IBD in apo (gray) and itaconate bound (green and blue) states. The largest structural rearrangement of each monomer is shown upon ligand binding. The N- and C-termini of the IBD are highlighted with the terminal residues in red and their associated displacement following ligand binding is labeled in angstroms. **c** Maxima values from emission peaks (monitored at 530 nm after excitation at

488 nm) of indicated itaconate biosensors were measured in the presence of 10 mM itaconate. These values were shown by heatmap after normalization to F0 (maxima values from emission peaks of the corresponding sensors in the absence of itaconate). Data are shown as mean ± SD (*n* = 3). **d** Schematic drawing of the itaconate biosensor BioITA. Fluorescent protein cpGFP was inserted between IBD and CENP-B. Itaconate binding changes conformation in protein to induce fluorescence. IBD itaconate-binding domain, cpGFP circularly permuted green fluorescent protein, CENP-B human centromere protein B. Source data are provided as a Source Data file.

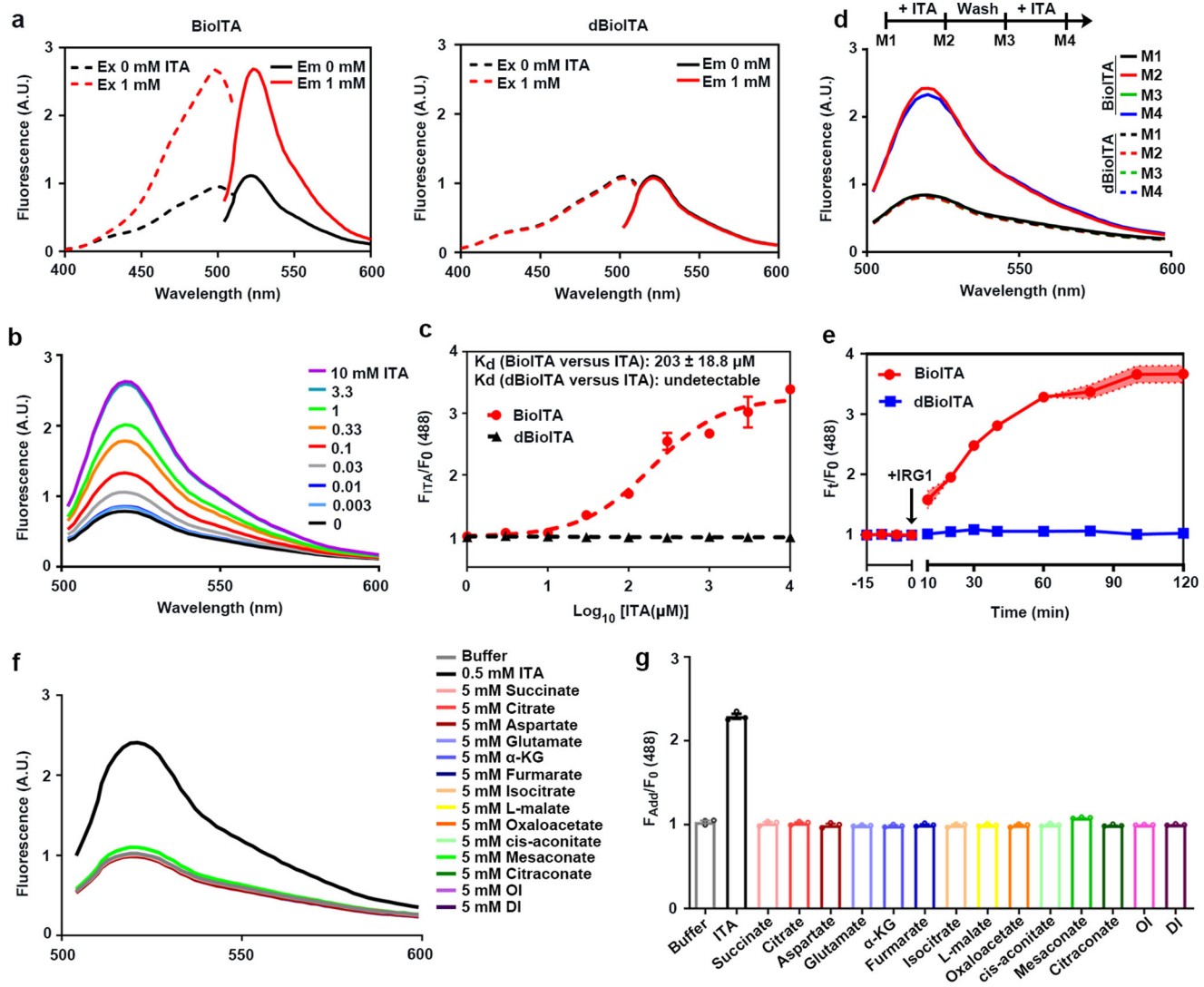

**Fig. 2 | Characterizing BioITA sensor. a** Fluorescence spectra of purified BioITA (left panel) and dBioITA (right panel) in the presence of 0 (black) or 1 mM itaconate (red). The excitation (Ex, dashed lines) spectrum recorded at an emission wavelength of 530 nm has a maximum at 488 nm; the emission (Em, solid lines) spectrum recorded at an excitation wavelength of 488 nm has a maximum at 530 nm. The experiments were repeated thrice independently with similar results. **b** Fluorescence emission scans (excitation at 488 nm) from BioITA at indicated itaconate concentrations. The scans using buffer serves as a control. The experiments were repeated thrice independently with similar results. **c** Maxima values from emission peaks (excitation at 488 nm) of BioITA and dBioITA versus indicated itaconate concentrations were plotted after normalization to F0 (without itaconate). The itaconate dissociation constant (Kd) of BioITA was determined as $203 \pm 18.8$ μM. The data indicated at different itaconate concentrations are shown as mean $\pm$ SD of three independent experiments. **d** Flowchart of fluorescence emission scan from BioITA. Fluorescence emission scans were performed before and after addition of itaconate (1 mM), and then the repeated fluorescence emission scans were performed following the itaconate elution from BioITA. Each time point of fluorescence emission scanning was indicated (upper panel). Fluorescence spectra of BioITA (solid lines) and dBioITA (dashed lines) were shown (lower panel). M1-4 measurement 1–4. The experiments were repeated thrice independently with similar results. **e** IRG1 (red line) increases BioITA's fluorescence (monitored at 530 nm after excitation at 488 nm). Data are shown as mean $\pm$ SD of three independent experiments. **f** Fluorescence emission scan (excitation at 488 nm) from BioITA in presence of the indicated metabolites. **g** Maxima values from emission peaks (excitation at 488 nm) of BioITA in presence of the indicated metabolites were shown after normalization to F0 (maxima value from emission peak of BioITA only). Data are shown as mean $\pm$ SD of three independent experiments. A.U., arbitrary unit (**a**, **b**, **d**, **f**). Source data are provided as a Source Data file.

488 nm upon itaconate addition (Fig. 1c). This chimera, termed BioITA (Fig. 1d and Supplementary Fig. 1g–i), was used for further characterization. In addition, a non-responsive control biosensor, designated dBioITA (dead BioITA), was engineered by incorporating the R62E mutation to abolish itaconate binding, as demonstrated by isothermal titration calorimetry assays (Supplementary Fig. 1j, k).

### Characterizing BioITA sensor

Next, we characterized the properties of the BioITA in detail. Purified BioITA and the control biosensor dBioITA (Supplementary Fig. 2a) had major excitation peaks at ~488 nm with emission peaks at ~530 nm

(Fig. 2a). Itaconate increased fluorescence (excitation at 488 nm) of BioITA in a dose dependent manner (Fig. 2b). Of note, itaconate binding did not affect fluorescence (excitation at 405 nm) of BioITA and dBioITA (Supplementary Fig. 2b), suggesting that fluorescence from excitation at 405 nm is able to serve as a loading control for protein levels of BioITA and dBioITA. In addition, we found that the apparent dissociation constant ($K_d$) of BioITA toward itaconate is $203 \pm 18.8$ μM (Fig. 2c), which falls into the range of intracellular itaconate concentration of activated macrophages[1].

To confirm the binding of itaconate to BioITA is reversible, we performed the fluorescence scanning from BioITA in the absence and

presence of itaconate and these measurements were repeated following itaconate elution. As expected, itaconate treatment resulted in obvious elevation of fluorescence emission from BioITA, but not the control biosensor dBioITA, and the repeated effect was observed following the itaconate elution from BioITA (Fig. 2d). In addition, BioITA was used to monitor the itaconate production by active IRG1 in real time. We observed that fluorescence emission from BioITA displayed a rapid increase upon addition of active IRG1 and achieved an equilibrium at about 60 min post active IRG1 addition (Fig. 2e and Supplementary Fig. 2a). In line with the BioITA-based measurement, biochemical analysis also demonstrated that itaconate production showed a rapid increase upon addition of active IRG1 and achieved a concentration (~2000 μM) close to the maximal working concentration of BioITA at 60 min post active IRG1 addition (Supplementary Fig. 2c). These results indicate that BioITA may rapidly and reliably detect itaconate within its range of working concentration in a reversible manner.

To determine BioITA specificity, we detected fluorescence in the presence of other intermediate metabolites in TCA cycle, cell permeable itaconate derivatives 4-octyl itaconate (OI) and dimethylitaconate (DI)[9], as well as itaconate isomers mesaconate and citraconate[30,31]. Elevated fluorescence from BioITA was observed only in presence of itaconate (Fig. 2f, g), suggesting that BioITA detects itaconate with a high specificity. Furthermore, we found that BioITA and dBioITA display similar responses to pH (Supplementary Fig. 2d), and the itaconate-dependent responses of BioITA were minimally affected from pH 6.5 to 8.0 (Supplementary Fig. 2e, f), suggesting that the pH effects can be corrected by measuring fluorescence of BioITA and dBioITA in parallel. In addition, fluorescence of BioITA is not significantly affected by temperature fluctuations between 22 and 37 °C (Supplementary Fig. 2g–i). Collectively, these data indicate that BioITA displays excellent selectivity and sensitivity toward itaconate, which makes it a promising tool for itaconate studies in living cells.

## Detecting itaconate in mitochondria and cytosol of living macrophages

To assess the suitability of BioITA in mitochondria and cytosol of living macrophages. We generated clonal RAW264.7 cells stably expressing BioITA or their corresponding dBioITA control with localization sequences targeting to the mitochondria and cytosol, respectively (Fig. 3a and Supplementary Fig. 1i). Mitochondrial (m) or cytosolic (c) BioITA, but not their corresponding control biosensor dBioITA, manifested marked increases of its fluorescence when cellular itaconate levels increased upon LPS stimulation (Fig. 3b, c). In addition, we found that the sensors without or with localization sequences display similar responses to itaconate in vitro (Supplementary Fig. 3a). Notably, expression of the sensors did not alter LPS-stimulated IRG1 expression (Supplementary Fig. 3b) and intracellular itaconate levels determined by liquid chromatography–mass spectrometry (LC-MS) analysis with a standard curve (Supplementary Fig. 3c, d). Taken together, these data demonstrate that BioITA can reliably detect itaconate generation with high sensitivity in mitochondria and cytosol of living macrophages.

Furthermore, to estimate concentrations of free intracellular itaconate, we accessed the fluorescence intensity of cytosolic BioITA in digitonin-permeabilized RAW264.7 cells using flow cytometry in the presence of external itaconate with determined concentrations. Digitonin-mediated cell permeabilization was assessed by influx assay of propidium iodide (PI) (Supplementary Fig. 3e). Based on the calibration curve generated by plotting the normalized mean fluorescence intensity (MFI) from cBioITA versus its corresponding equilibrated itaconate concentration (Supplementary Fig. 3f), we estimated that the concentration of free itaconate was 551 μM [95% confidence interval (CI): 457–645 μM] in mitochondria and 1757 μM (95% CI: 1269–2245 μM) in cytosol of non-permeabilized RAW264.7 cells stimulated with LPS for 12 h (Supplementary Fig. 3g).

We next used BioITA to detect the effect of treatments with exogenous itaconate or cell permeable itaconate derivatives on intracellular itaconate level of macrophages. Intriguingly, cBioITA-expressing, but not the control biosensor cdBioITA-expressing, RAW264.7 cells exhibited marked increase of fluorescence upon treatment with unmodified itaconate (Supplementary Fig. 4a). In contrast, moderate increase of cBioITA's fluorescence were observed in RAW264.7 cells upon treatment with OI or DI (Supplementary Fig. 4a). These data imply that exogenous itaconate, as well as derivatized itaconate, is capable of elevating intracellular itaconate level of macrophages.

To provide evidence supporting that the binding of itaconate to BioITA is reversible in live cells, mouse IRG1 protein was exogenously expressed under the control of a tetracycline-inducible promoter in RAW264.7 cells stably expressing mitochondria- or cytosol-localized BioITA. Immunoblotting analysis indicated that pulse treatment with doxycycline for 30 min induced a transient expression of IRG1 during 4–8 h post doxycycline treatment (Supplementary Fig. 4b). As expected, fluorescence intensity of mitochondrial or cytosolic BioITA in RAW264.7 cells exhibited a transient elevation during the time period of IRG1 expression (Supplementary Fig. 4c, d). In line with this finding, intracellular itaconate level exhibited a transient elevation during the time period of IRG1 expression, as evidenced by the biochemical assay (Supplementary Fig. 4e). Taken together, our data demonstrate that BioITA is able to detect fluctuations of mitochondrial and cytosolic itaconate in living macrophages, thereby supporting the binding of itaconate to BioITA is reversible in live cells.

Itaconate generated by IRG1 in the mitochondrial matrix is predominantly transported across the mitochondrial inner membrane by the 2-oxoglutarate carrier (OGC)[5]. To examine the role of OGC in maintenance of compartmentalized itaconate pools, we used the clustered regularly interspaced short palindromic repeats (CRISPR)/Cas9 genome editing technology to knockout (KO) endogenous *Ogc* in RAW264.7 cells (Supplementary Fig. 4f). As expected, *Ogc* KO further increased mitochondrial BioITA's fluorescence (Fig. 3d), conversely, decreased cytosolic BioITA's fluorescence (Fig. 3e) upon LPS stimulation. As a control, minimal changes in fluorescence were observed in mitochondrial or cytosolic dBioITA-expressing RAW264.7 cells without or with *Ogc* KO upon LPS stimulation (Fig. 3d, e). These data imply that the maintenance of cytosolic itaconate pool depends on transportation of itaconate from mitochondria.

## BioITA reports itaconate metabolism inside living macrophages

In mammalian cells, itaconate is synthesized by IRG1 and catabolized into itaconyl-CoA, citramalyl-CoA, and, finally, pyruvate and acetyl-CoA via sequential reactions catalyzed by succinyl-CoA ligase, methylglutaconyl-CoA hydratase and CLYBL in mitochondria[12] (Fig. 4a). As shown by BioITA's fluorescence, LPS stimulation effectively increased mitochondrial and cytosolic itaconate levels in a time-dependent manner in wild-type (WT), but not *Irg1*-KO, RAW264.7 cells (Fig. 4b, c and Supplementary Fig. 5a). Consistent with this BioITA-based measurement, the results from biochemical assays also showed that intracellular itaconate level increased in a time-dependent manner upon LPS stimulation in WT, but not *Irg1*-KO, RAW264.7 cells (Supplementary Fig. 5b). Moreover, KO of *Suclg1* (Supplementary Fig. 5c), the gene encodes the subunit of succinyl-CoA ligase complex which metabolizes itaconate into its corresponding CoA ester itaconyl-CoA, further increased BioITA's fluorescence upon LPS stimulation (Fig. 4d, e). Consistently, the results obtained from biochemical analyses also indicated that intracellular itaconate levels of *Suclg1*-KO RAW264.7 cells were further increased compared to that of WT cells upon LPS stimulation (Supplementary Fig. 5d). Overall, these data demonstrate that BioITA correctly reports the increased levels of mitochondrial and cytosolic itaconate in LPS-activated macrophages and reveals the specific role of IRG1 and succinyl-CoA ligase as valuable targets for interfering itaconate metabolism.

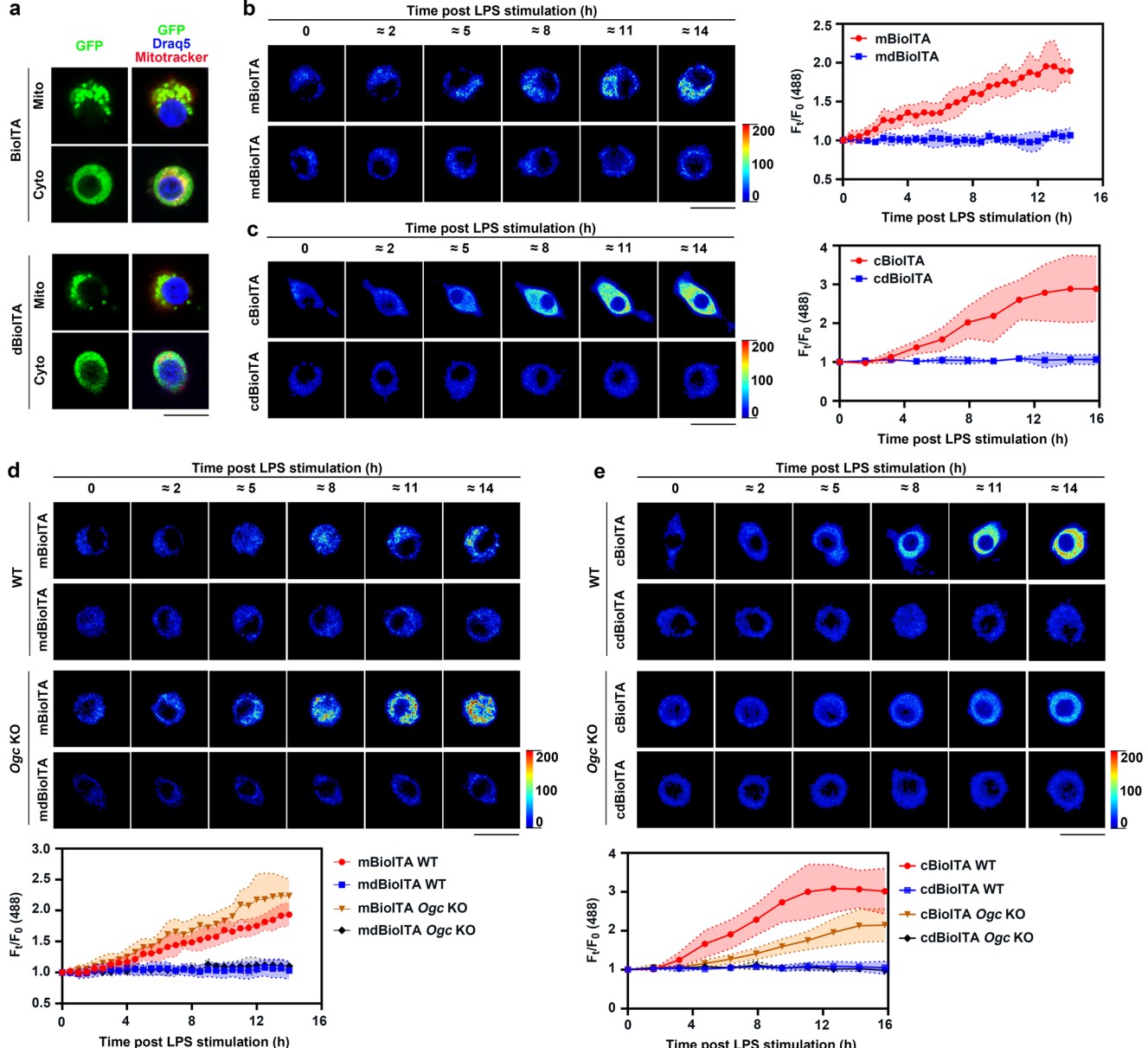

**Fig. 3 | BioITA images itaconate levels in mitochondria and cytosol of living macrophages. a** Representative live images of RAW264.7 cells stably expressing BioITA or dBioITA containing mitochondrial (mBioITA and mdBioITA) or cytosolic (cBioITA and cdBioITA) localization sequence. dBioITA, dead BioITA; mBioITA, mitochondrial BioITA; cBioITA, cytosolic BioITA; mdBioITA, mitochondrial dBioITA; cdBioITA, cytosolic dBioITA. Mitochondrial marker Mitotracker CMXRos, red; BioITA, green; nuclear marker Draq5, blue. Scale bar, 20 μm. The experiments were repeated three times independently with similar results. **b, c** RAW264.7 cells stably expressing mitochondrial (**b**) or cytosolic (**c**) BioITA was stimulated with or without LPS (10 ng ml⁻¹). Representative pseudo-color images were captured (**b, c**, left panels) and fluorescence intensity was quantitated at indicated time points post LPS stimulation (**b, c**, right panels). Mitochondrial or cytosolic dBioITA served as a control. Scale bar, 20 μm (**b, c**, left panels). Data are shown as mean ± SD (*n* = 8) (**b, c**, right panels). **d, e** Representative pseudo-color images of mitochondrial (**d**) or cytosolic (**e**) BioITA in RAW264.7 cells without or with *Ogc* KO were captured at indicated time points post LPS (10 ng ml⁻¹) stimulation (**d, e**, upper panels). Time-dependent fluorescence intensity of BioITAs or dBioITA in individual cell after LPS stimulation was quantitated (**d, e**, lower panels). Mitochondrial or cytosolic dBioITA served as a control. Scale bar, 20 μm (**d, e**, upper panels). Data are shown as mean ± SD (*n* = 8) (**d, e**, lower panels). Dotted lines indicated the mean ± SD (**b, c**, right panels; **d, e**, lower panels). Source data are provided as a Source Data file.

## BioITA reports itaconate dynamics upon STING activation

To further test whether BioITA can report itaconate production in response to STING activation, we performed imaging analyses of mitochondrial and cytosolic BioITA-expressing RAW264.7 cells upon stimulation of STING agonist 2'3'-cGAMP. Indeed, 2'3'-cGAMP stimulation significantly elevated *Irg1* mRNA level of RAW264.7 cells in a time-dependent manner although this effect was delayed compared to LPS stimulation (Supplementary Fig. 6a). As expected, 2'3'-cGAMP stimulation significantly increased mitochondrial (Fig. 5a) and

cytosolic (Fig. 5b) itaconate levels in a time-dependent manner as shown by the fluorescence of BioITA, whereas this effect was abolished by *Sting1* KO (Fig. 5a, b and Supplementary Fig. 6b). Consistently, the results obtained from biochemical analyses also demonstrated that 2'3'-cGAMP stimulation induced IRG1 expression (Supplementary Fig. 6c) and elevated intracellular itaconate levels (Supplementary Fig. 6d) in WT, but not *Sting1*-KO, RAW264.7 cells. Collectively, these data show that BioITA can reliably report the increase of intracellular itaconate levels with high sensitivity during STING activation.

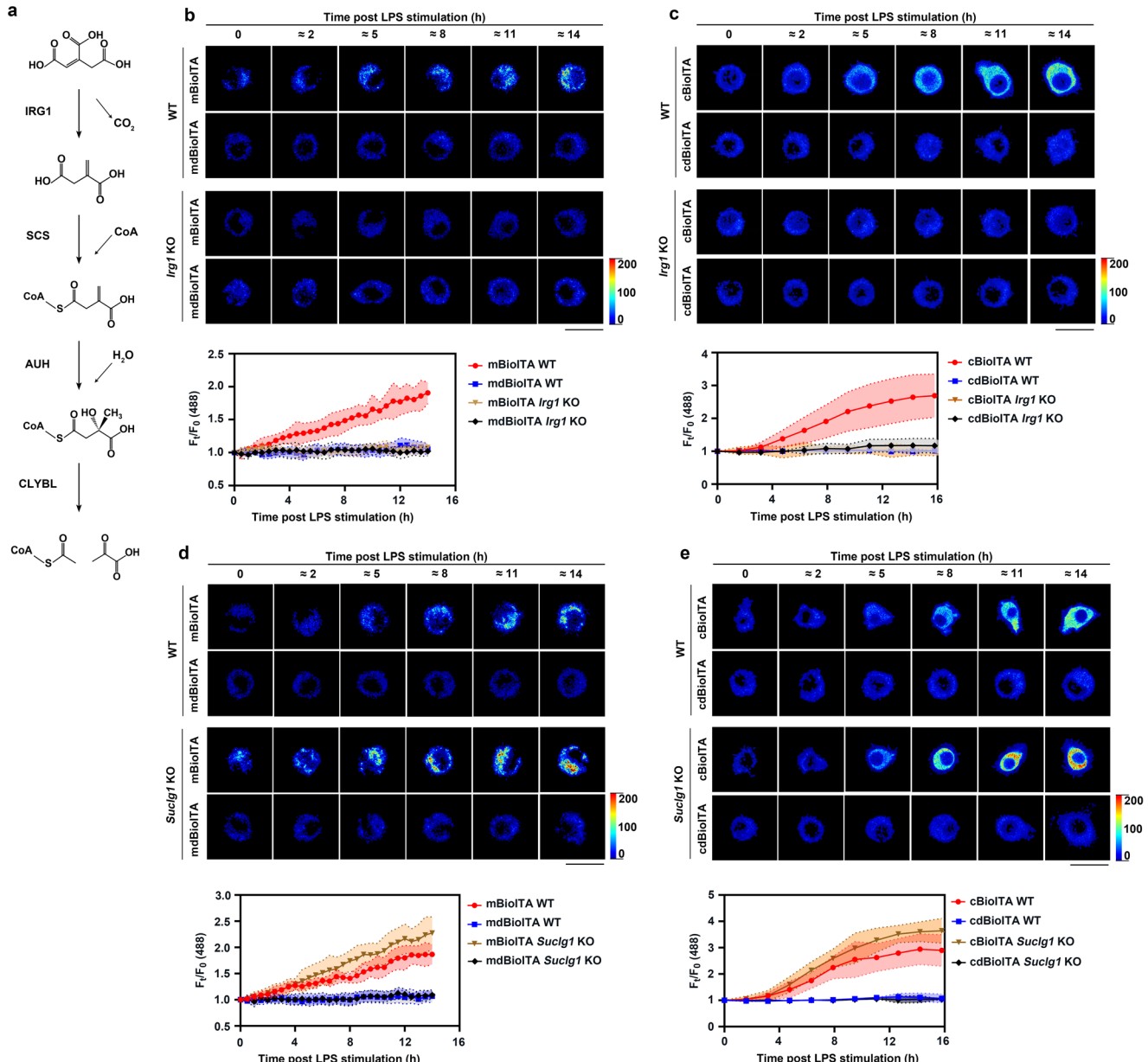

**Fig. 4 | BioITA reports itaconate metabolism in living macrophages. a** The central nodes for cellular itaconate metabolism. SCS, succinyl-CoA ligase; AUH, methylglutaconyl-CoA hydratase; CLYBL, citramalyl-CoA lyase. **b, c** Representative pseudo-color images of mitochondrial (**b**) or cytosolic (**c**) BioITA in RAW264.7 cells with or without *Irg1* KO were captured at different time points after LPS (10 ng ml⁻¹) stimulation (**b, c**, upper panels). Time-dependent fluorescence intensity of BioITAs or dBioITA in individual cell after LPS stimulation was quantitated (**b, c**, lower panels). Mitochondrial or cytosolic dBioITA served as a control. Scale bar, 20 µm (**b, c**, upper panels). Data are shown as mean ± SD (*n* = 8) (**b, c**, lower panels).

**d, e** Representative pseudo-color images of mitochondrial (**d**) or cytosolic (**e**) BioITA in RAW264.7 cells with or without *Suclg1* KO were captured at different time points after LPS (10 ng ml⁻¹) stimulation (**d, e**, upper panels). Time-dependent fluorescence intensity of BioITAs or dBioITA in individual cell after LPS stimulation was quantitated (**d, e**, lower panels). Mitochondrial or cytosolic dBioITA served as a control. Scale bar, 20 µm (**d, e**, upper panels). Data are shown as mean ± SD (*n* = 8) (**d, e**, lower panels). Dotted lines indicated the mean ± SD (**b–e**, lower panels). Source data are provided as a Source Data file.

## BioITA detects itaconate in activated macrophages derived from LPS-injected mice

To investigate itaconate dynamics during inflammation ex vivo, we applied the BioITA to a mouse model of LPS-induced sepsis. We injected the mice with an adeno-associated virus (AAV) coding cytosolic BioITA (cBioITA) or the control biosensor cdBioITA under the control of a macrophage-specific promoter CD68/150[32] (Fig. 6a). We observed a significantly increase of itaconate level in macrophages freshly isolated from LPS-injected mice compared with that from mice without this treatment, as monitored with live flow cytometry (Fig. 6b and Supplementary Fig. 7a, b) and microscopic imaging (Fig. 6c). As a control, minimal changes in fluorescence were observed in cdBioITA-expressing macrophages freshly isolated from these mice (Fig. 6b, c and Supplementary Fig. 7b). In line with these findings, induced expression of IRG1 (Supplementary Fig. 7c) and marked increase of intracellular itaconate levels (Supplementary Fig. 7d) were detected by biochemical assays in macrophages freshly isolated from LPS-injected mice. In summary, these data indicate that BioITA is able to report the itaconate dynamics in macrophages ex vivo.

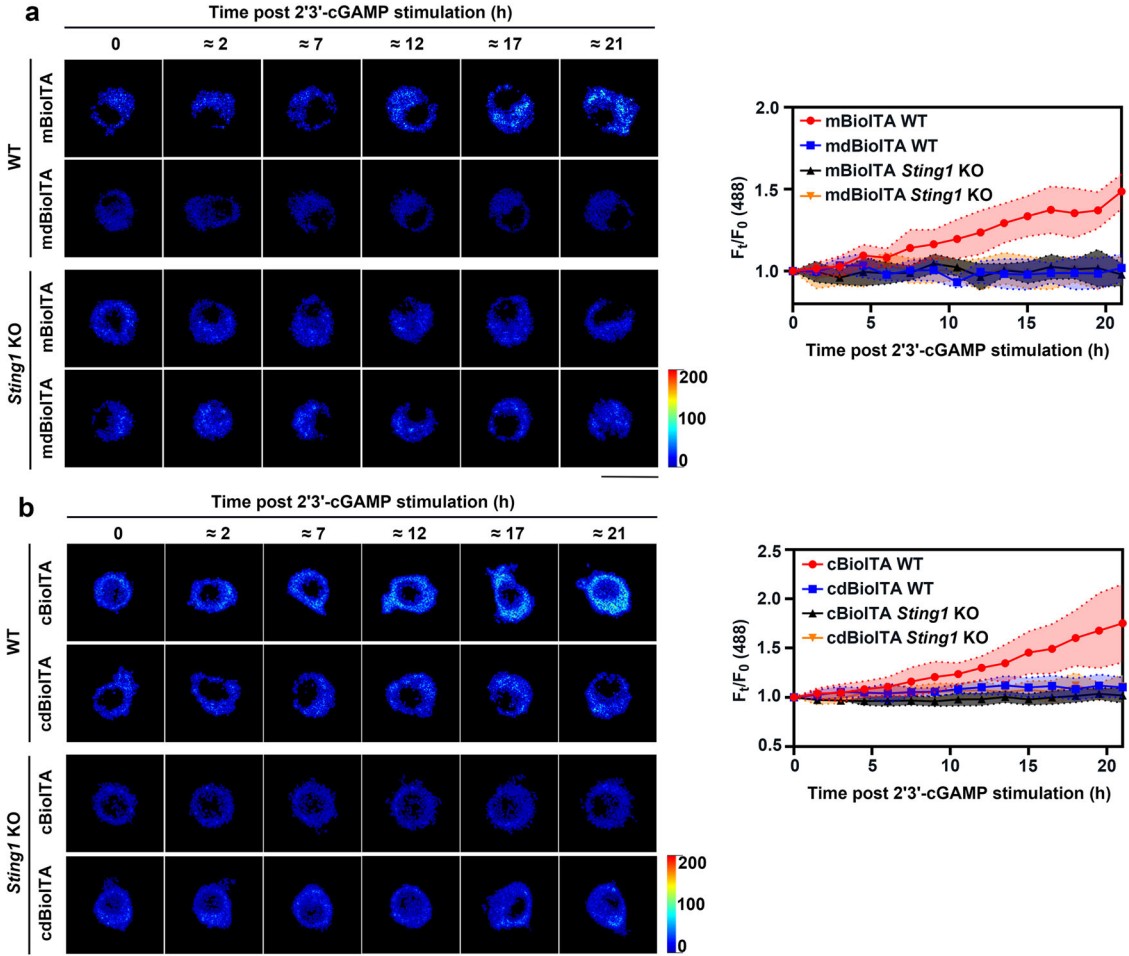

**Fig. 5 | BioITA reports itaconate dynamics upon STING activation.**
**a**, **b** Representative pseudo-color images of mitochondrial (**a**) or cytosolic (**b**) BioITA in RAW264.7 cells without or with *Sting1* KO were captured at indicated time points post 2'3'-cGAMP (10 μg ml⁻¹) stimulation (**a**, **b**, left panels). Time-dependent fluorescence intensity of BioITAs or dBioITA in individual cell after 2'3'-cGAMP stimulation was quantitated (**a**, **b**, right panels). Mitochondrial or cytosolic dBioITA served as a control. Scale bar, 20 μm (**a**, **b**, left panels). Data are shown as mean ± SD (*n* = 8) (**a**, **b**, right panels). Dotted lines indicated the mean ± SD (**a**, **b**, right panels). Source data are provided as a Source Data file.

## Discussion

In this study, we have developed and validated a genetically encoded fluorescent itaconate probe, BioITA biosensor. BioITA has the sensitivity, ligand specificity, and kinetics suitable for monitoring itaconate dynamics in vitro and in living macrophages, providing a reliable tool for monitoring itaconate levels with temporal and spatial resolution.

It has been known that itaconate levels increase in LPS-stimulated macrophages; however, all of these results are obtained from biochemical analyses at cell pool level. By using BioITA, we can detect itaconate within the range of physiological concentrations in live individual cells. In addition, we also show that, as a genetically encodable protein, BioITA senses itaconate in distinct cellular compartments, such as mitochondria and cytosol, through fusion with distinct localization tags. Thus, BioITA biosensor provides a powerful tool for subcellular-resolution imaging itaconate within living cells.

Itaconate is identified as an anti-inflammatory metabolite and regulator of type I IFNs in the context of inflammation and immunity[5]. A physiological pool of itaconate is maintained in macrophages to protect against LPS-induced damages during inflammation. The changes in BioITA's fluorescence intensity actually reflect intracellular itaconate fluctuations under inflammatory circumstances. As expected, the BioITA responds well to itaconate production upon inflammatory activation, consistent with the biochemical analysis of

intracellular itaconate content. Moreover, BioITA also correctly reports the decreases or increases of itaconate when metabolism of itaconate is modulated by knocking out its biosynthesis gene *Irg1* or catabolism gene *Suclg1*. Hitherto, BioITA is the only sensor that allows sensitive, specific, and real-time readout of itaconate metabolism in living macrophages. BioITA, which is effective in interrogating itaconate's roles in biologically and medically important processes, should advance our understanding to the pathogenesis of various inflammatory diseases. We therefore endorse the use of this tool to further investigate the unknown biological functions of itaconate, which may have important therapeutic or clinical implications.

## Method

### Ethical statement

All research complied with relevant ethical regulations (Protocol # SYXK2019030) and were approved by the Institutional Animal Care and Use Committee of the Institute of Biophysics, Chinese Academy of Sciences.

### Materials

Rabbit polyclonal antibodies recognizing mouse IRG1 (#17805, 1:2000 for immunoblotting), rabbit monoclonal antibody recognizing STING (#13647, 1:2000 for immunoblotting), TBK1 (#38066S, 1:2000 for immunoblotting), p-TBK1 (#5483S, 1:2000 for immunoblotting), and

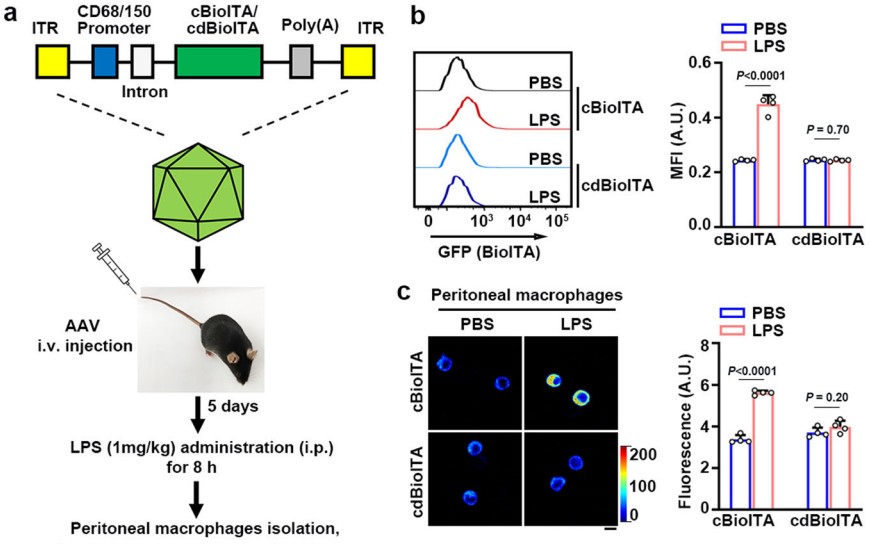

**Fig. 6 | BioITA detects itaconate in activated macrophages derived from LPS-injected mice. a–c** Schematic illustration of intravenous injection of AAV ($1 \times 10^{11}$ viral genomes in 100 μl PBS) encoding cBioITA or the control biosensor cdBioITA under the control of a macrophage-specific promoter CD68/150 into 6-week-old female C57BL/6J mice ($n = 4$). The mice were intraperitoneally injected without or with LPS (1 mg kg⁻¹) on day 5 post AAV injection and peritoneal macrophages were isolated 8 h after LPS injection. *P* values were determined by the two-tailed Student's *t*-test. ITR inverted terminal repeat, cBioITA cytosolic BioITA, cdBioITA cytosolic dBioITA. **b** The mean fluorescence intensity (MFI) measured with flow cytometry (excitation at 488 nm, emission at 525 nm) from cBioITA and cdBioITA (left panel) was quantitated and the data are presented as mean ± SEM (right panel). **c** Representative pseudo-color images of cBioITA and cdBioITA in peritoneal macrophages were shown (left panel). The immunofluorescence intensity in 30 cells from each mouse was quantitated and the data are shown as mean ± SEM (right panel). Scale bar, 20 μm. A.U. arbitrary unit (**b**, **c**). Source data are provided as a Source Data file.

MitoTracker (#8778) were purchased from Cell Signaling Technology. Rabbit polyclonal antibodies recognizing OGC (#K006830P, 1:500 for immunoblotting) was purchased from Beijing Solarbio Science & Technology. Rabbit polyclonal antibodies recognizing SUCLG1 (#14923-1-AP, 1:500 for immunoblotting) was purchased from Proteintech. Mouse monoclonal antibody recognizing β-actin (#T0022-HRP, 1:2000 for immunoblotting) was purchased from Affinity Biosciences. PE/Cyanine7 anti-mouse F4/80 (#123113, 1:500 for immunofluorescence) and Pacific blue anti-mouse/human CD11b (#101223, 1:500 for immunofluorescence) were purchased from Biolegend. Rabbit polyclonal antibodies recognizing GFP (#G1544, 1:2000 for immunoblotting), itaconate (#I29204), dithiothreitol (#D9779), phenylmethanesulfonyl fluoride (PMSF) (#P7626), LPS (#L2880), IPTG (#I6758), 4',6-diamidino-2-phenylindole (DAPI) (#D8417), selenomethionine (#1611955), succinate (#224731), citrate (#C3674), aspartate (#A8949), glutamate (#G1251), α-KG (#75890), furmarate (#F1506), isocitrate (#I1252), L-malate (#02288), oxaloacetate (#O4126), cis-aconitate (#A3412), methanol (#900641), chloroform (#151823), BL21(DE3) pLysS (#C606010), B834(DE3) (#69041), glycerol (#G5516), propidium iodide (#81845), 4-octyl itaconate (#SML2338), dimethyl-itaconate (#109533), digitonin (#11024-24-1), mesaconate (#131040), citraconate (#C82604), and hygromycin B (#V900372) were purchased from Sigma-Aldrich. Horseradish peroxidase (HRP)-conjugated goat anti-mouse (#G-21040, 1:2000 for immunoblotting) and goat anti-rabbit (#G-21234, 1:2000 for immunoblotting) secondary antibodies were purchased from Thermo Fisher Scientific. Lipofectamine 3000 transfection reagent (#L3000015) was purchased from Invitrogen. 2'3'-cGAMP (#tlrl-nacga23) was purchased from InvivoGen. HiLoad 16/600 Superdex 200 column was purchased from GE healthcare. Mouse antibody recognizing α-tubulin (#sc-23948, 1:2000 for immunoblotting), β-mercaptoethanol (#sc-202966), puromycin (#sc-108071), doxycycline (#sc-204734B) and polybrene (#sc-134220) were purchased from Santa Cruz Biotechnology. Crystal screening kits: Index (#HR2-144), Crystal screen (#HR2-110, #HR2-112), PEGRx (#HR2-082, #HR2-084), PEG/Ion (#HR2-126, #HR2-98) were purchased from Hampton Research.

## Cell lines and cell culture conditions

RAW264.7 cells (ATCC: Catalog # TIB-71) were maintained in Dulbecco's modified Eagle's medium (DMEM) supplemented with 10% fetal bovine serum (HyClone), 1% penicillin-streptomycin at 37 °C with 5% CO₂. The medium was changed daily and the cell line were routinely tested for mycoplasma contamination. During imaging, cells were maintained in phenol red free DMEM (Invitrogen) supplemented with 10% fetal bovine serum, 1% penicillin-streptomycin at 37 °C with 5% CO₂.

## Construction of expression vectors

All PCR primers used for molecular cloning are listed in Supplementary Table 2 and plasmids are listed in Supplementary Table 3. DNA fragments encoding prototype version of BioITA was synthesized by Genewiz company and ligated into XhoI- and BamHI-digested pET28a vector. To construct vectors expressing linker version of BioITA, DNA sequences encoding various linkers (connecting C-terminus of IBD to N-terminus of cpGFP and C-terminus of cpGFP to N-terminus cpGFP of CENP-B) were inserted into expression cassette of prototype version of BioITA via Gibson Assembly using a NEBuilder HiFi DNA Assembly Kit (New England Biolabs #E2621) according to the manufacturer's instructions. The non-responsive control biosensor dBioITA (BioITA R62E) was constructed using a QuikChange site-directed mutagenesis kit (Stratagene) and primers 1 and 2. To construct pET28a-based vectors expressing BioITA with a leader sequence, the gene encoding BioITA including cytosolic or mitochondrial localization sequence at N-terminus was PCR-amplified from pET28a-BioITA using primers 3-5, and then ligated into BamHI- and XhoI-digested pET28a vector. To construct lentiviral vector expressing cytosol- or mitochondria-localized BioITA, the gene encoding BioITA tagged with N-terminal cytosolic or mitochondrial localization sequence was subcloned into BglII- and SalI-digested pLenti lentiviral vector (Thermo Fisher Scientific) using primers 6-8. Likewise, the AAV vector expressing cytosol-localized BioITA (cBioITA) was constructed by subcloning the DNA fragment containing expression cassette of cBioITA into EcoRI- and

BamHI-digested AAV2 vector (Clontech Laboratories) using primers 9 and 10.

For expression of itaconate-binding domain (IBD) (ItcR truncation of residues 80–292), DNA sequences encoding IBD were amplified by PCR using primers 11 and 12, and then ligated into BamHI- and XhoI-digested pET28a vector. For expression of recombinant human IRG1 protein, the genes encoding human IRG1 PCR-amplified from pcDNA3.1-Flag-IRG1 vector was subcloned into BamHI- and XhoI-digested pET28a vector using primers 13 and 14. To construct the vector for inducible expression of mouse IRG1 in RAW264.7 cells, the genes encoding mouse IRG1 was subcloned into EcoRI- and BamHI-digested pLVX-TetOne-Puro lentiviral vector using primers 15 and 16.

## Protein expression and purification
BL21 (DE3) pLysS cells transformed with pET28a-based plasmids expressing various recombinant proteins, including different versions of BioITAs, IBD, and human IRG, were cultured in 250 ml LB medium and treated with 100 µM IPTG for 20 h at 18 °C before lysis by sonication. For production of selenomethionine-labeled IBD for crystallography, pET-28 plasmid expressing 8 × His-tagged and TEV-cleavage-site-fused IBD was transformed into B834 (DE3) cells and grown for 16 h at 37 °C in Overnight Express Autoinduction System 2 (Novagen, # 71366) medium containing 125 µg ml$^{-1}$ selenomethionine. Cells were harvested by centrifugation and then lysed with BugBuster reagent. The cell lysates were sonicated and centrifuged at 12,000 × g for 10 min. Soluble cell extracts were collected for downstream purification.

Purification of His-tagged proteins was performed by loading cell extracts onto a nickel-nitrilotriacetic acid column (GE Healthcare Life Sciences) followed by washing with five column volumes of 20 mM imidazole and subsequent elution with 300 mM imidazole. The eluted samples were dialyzed overnight against PBS and His tag of the proteins was removed by digestion with 50 µg ml$^{-1}$ his-tagged TEV protease for 4 h at 30 °C. The cleavage mixtures were loaded onto a nickel-nitrilotriacetic acid column to remove the undigested target proteins and His-tagged TEV protease. The flow-through fractions containing the His-tag free target proteins were further purified by gel filtration using a HiLoad 16/600 Superdex 200 column (GE Healthcare). Purity of the proteins was examined by SDS-PAGE followed by colloidal *Coomassie* blue (G-250) staining. The purified proteins were concentrated to 5–40 mg ml$^{-1}$ for the following structural and biochemical studies.

## Isothermal titration calorimetry (ITC) assay
The purified biosensor proteins and itaconate were diluted into the same buffer (20 mM HEPES pH [7.4], 150 mM NaCl, 1 mM DTT). Binding affinity data were obtained by using MicroCal ITC-200 (GE Healthcare) at 25 °C. Sensor protein was thermostated in the sample cell, and itaconate was then injected stepwise over 20 injections with 120 s space apart. The concentrations of sensor protein and itaconate were determined as 50 µM and 5 mM, respectively. Data were analyzed by using Origin 7.0 software (Origin Laboratory).

## Multiangle light scattering (MALS) analysis
Molecular mass of purified ItcR itaconate-binding domain (IBD) was analyzed by an analytical light scattering instrument consisting of a 1260 infinity II LC system (Agilent Technology), a DAWN Heleos-II multiangle light scattering detector (Wyatt Technology), and an Optilab T-rEX refractive index detector (Wyatt Technology). Briefly, protein sample (100 µl) was loaded onto a HiLoad 10/300 Superdex 200 column (GE Healthcare) connecting to 1260 infinity II LC system and then eluted at a flow rate of 0.4 ml min$^{-1}$. Column effluent was monitored simultaneously with three detectors for 280 nm ultraviolet light absorption, light scattering and refractive index. Data were analyzed by using ASTRA software (Wyatt Technology) to determine molecular mass of the protein.

## Crystallization
For crystallization of ItcR IBD in complex with itaconate, ItcR IBD (5 mg ml$^{-1}$) without or with selenomethionine (SeMet) incorporation was mixed with itaconate at a molar ratio of 1:100. The mixture was incubated at 20 °C for 30 min. The hanging-drop vapor diffusion method was performed by mixing equal volume (1 µl:1 µl) of protein sample and reservoir solution (100 mM Bis-Tris, pH [6.5], 25% v/v PEG300) at 20 °C. For crystallization of apo ItcR IBD, the crystals were grown by mixing 1 µl of the protein sample (5 mg ml$^{-1}$) and 1 µl of the reservoir solution (200 mM sodium citrate tribasic dihydrate, pH [8.3], 20% w/v PEG3350) at 20 °C. Crystals were cryoprotected using the reservoir solution supplemented with 20% glycerol and flash-cooled in liquid nitrogen immediately for diffraction data collection.

## Structure determination
All the diffraction data sets were collected at BL17U1 and BL18U1 beamlines at the Shanghai Synchrotron Radiation Facility (SSRF). Data were indexed, integrated, and scaled with the XDS program suite. The crystal structures of SeMet-IBD in complex with itaconate were solved by multiple-wavelength anomalous diffraction method using the program AutoSol in PHENIX. The structure of apo ItcR IBD and ItcR IBD in complex with itaconate was solved by molecular replacement using the SeMet-IBD structure as the searching model. Model building and refinement for all the structures were carried out using the programs COOT and PHENIX, respectively. The statistics of the diffraction data and refinement are shown in Supplementary Table 1. Structure figures were prepared by using the Pymol software.

## Fluorescence spectroscopy
The recombinant fluorescent biosensor proteins dissolved in buffer (100 mM HEPES, 150 mM NaCl) were placed into a black 96-well flat bottom plate. Excitation and emission spectra of the biosensors were measured by using a Synergy H1 spectrofluorometer (BioTek) at the set temperatures in presence of different concentrations of itaconate. Excitation spectra were monitored at 530 nm and emission spectra were measured by excitation at 488 or 405 nm. Readings were taken every 2 nm with an integration time of 1 s. Unless stated, the purified biosensor protein concentration used was 250 nM and the fluorescent biosensors were monitored under the condition of temperature 25 °C and pH 7.4.

## IRG1 enzyme activity measurement
*Cis*-aconitate decarboxylase activity of IRG1 was measured by adding 10 µg purified human IRG1 into 100 µl reaction mixture (25 mM HEPES, pH [7.4], 10 mM *cis*-aconitate) in presence of 250 nM sensor (BioITA or dBioITA) at 30 °C. The sensor's fluorescence monitored at 530 nm after excitation at 488 nm was recorded every 10–20 min.

## Metabolite profiling with LC−MS
For measurement of intracellular itaconate levels, cells grown in 60-mm dish or isolated peritoneal macrophages (~1 × 10$^6$) were washed twice with PBS and then quenched with 1 ml −20 °C methanol. An equal volume of 4 °C water was added into each dish, and the samples were collected with a cell scraper and transferred into tubes containing 1 ml −20 °C chloroform. For measurement of itaconate production catalyzed by IRG1, the samples collected at different reaction time points were mixed with 1 ml −20 °C methanol followed by addition of an equal volume of 4 °C water, and then the samples were transferred into tubes containing 1 ml −20 °C chloroform. The extracts were vortexed at 1400 rpm for 20 min at 4 °C followed by centrifuge at 16,000 × g for 5 min at 4 °C. The upper aqueous phase was transferred into a specific GC glass vial followed by evaporation under vacuum at −4 °C using a refrigerated CentriVap Concentrator (Labconco). Dried extracts were dissolved in 2 ml methanol, and chromatographically separated using an Agilent ZORBAX 300SB-C8 column (3.5 µm, 2.1 mm × 50 mm) at 25 °C. 0.1% formic acid in water (A) and 0.1% formic acid in acetonitrile

(B) were used as mobile phase. The column was run at a flow rate of 0.2 ml min⁻¹ following the program: 99% A and 1% B during 0–7 min, 85% A and 15% B during 7–13 min, and 99% A and 1% B during 13–15 min. HRAM data were obtained from a liquid chromatography/Quadrupole-TOF mass spectrometer (Agilent 6530) and mass spectrometer was operated in a negative mode. The absolute concentration of itaconate was calculated according to the standard curve of itaconate.

### Immunoblotting

One million of cells grown in 60-mm dish or isolated peritoneal macrophages were washed twice with PBS and then lysed with 0.5 ml modified lysis buffer (50 mM Tris-HCl pH [7.5], 0.1% SDS, 1% Triton X-100, 150 mM NaCl, 1 mM dithiothreitol, 0.5 mM EDTA, 100 μM PMSF, 100 μM leupeptin). Cell debris were removed by centrifugation at 13,400 × g for 10 min at 4 °C. The lysate samples were resolved on 8%, 10% or 12% polyacrylamide minigels (Bio-Rad) and transferred onto PVDF membrane (GE Healthcare Life Sciences, PA) by a wet transfer system. The membranes were probed with primary and then HRP-conjugated secondary antibodies. Immunoblots were detected by using SuperSignal West Pico chemiluminescent substrate (Thermo Fisher Scientific, MA) and visualized by a ChemiScope 6000 Exp instrument. Photoshop CS6 (Adobe Inc.) was used to crop images from unprocessed images.

### Lentiviral production and generation of stable cell lines

The DNA sequences encoding itaconate biosensors fused with different eukaryotic subcellular localization signal peptides were subcloned into a pLenti lentiviral expression vector (Thermo Fisher Scientific, MA) with selectable marker of hygromycin. Lentiviruses were produced by co-transfecting 293FT cells with lentiviral plasmid containing coding sequence of itaconate biosensor, and two packaging plasmids pMD2.G (#12259) and psPAX2 (AddGene #12260). At 72 h post plasmid transfection, infectious lentiviruses were harvested and cell debris was removed by centrifuging the culture supernatants at 300 × g for 5 min followed by filtrating with a 0.45 μm filter (Millipore, MA). RAW264.7 cells were infected with lentiviruses at a multiplicity of infection [MOI] of 1 in presence of 10 μg ml⁻¹ polybrene, and then selected by 200 μg ml⁻¹ hygromycin for 14 days. The efficiency of ectopic expression was evaluated by immunoblotting and observed under a confocal microscope.

For knockout (KO) of *Irg1*, *Ogc*, and *Sting1*, guide RNAs (gRNAs) targeting *Irg1*, *Ogc*, and *Sting1* were designed using Cas9 target design online tool (http://www.genome-engineering.org). The non-targeting control guide RNA or three different promising guide RNAs for each gene were selected and subcloned into the lentiCRISPRv2 lentiviral vector with selectable marker of puromycin. The presence of the guide RNA was confirmed by Sanger sequencing. Lentiviruses were produced by the pMD2.G/psPAX2 packaging system. Infections of biosensor-expressing RAW264.7 cells were performed in 6-well plates at a MOI of 1 in presence of 10 μg ml⁻¹ polybrene, and uninfected cells were killed by incubation with 1 μg ml⁻¹ of puromycin for 7 days. KO efficiency of the genes was evaluated by immunoblotting. Target sequences of gRNAs for each gene are listed in Supplementary Table 4.

For inducible expression of mouse IRG1, the DNA sequence encoding *Irg1* was subcloned into a pLVX-TetOne lentiviral expression vector (#631849, Takara Bio) with selectable marker of puromycin. Lentiviruses were packaged according to the manufacturer's instructions and biosensor-expressing RAW264.7 cells were infected with these virions at a MOI of 1 in presence of 10 μg ml⁻¹ polybrene followed by incubation with 1 μg ml⁻¹ of puromycin for 7 days. Inducible expression of mouse IRG1 was evaluated by immunoblotting post incubation of the cells with doxycycline (1 μg ml⁻¹).

### Fluorescence imaging of live cells

For imaging of sensor-expressing RAW264.7 cells or sorted peritoneal macrophages, cells were plated on 35-mm glass-bottom dish. BioITA

and the control biosensor dBioITA were expressed in subcellular compartments by tagging with organelle-specific signal peptides. Images were acquired using an inverted Zeiss LSM980 confocal microscope system equipped with an Airyscan 2 detector and a live-cell imaging chamber supplied with 37 °C temperature and 5% CO₂. Plan Apo 40×/1.30 NA oil objective was utilized. Cells were excited at 488 nm and monitored with emission 525 nm. Z-stack images were constructed by combining 7–8 cell images taken at different focal distances with a range of 20 μm. Cell images were acquired one time before stimulant addition (set as F0) and dozens of times (set as Ft) post stimulant addition. For each condition, at least 5 fields containing 50–100 cells were used for imaging. Pixel intensity was quantitated by using Imairs software and background was subtracted from each image.

### RNA isolation and quantitative PCR

Total RNA was extracted from cultured RAW264.7 cells using TRIzol reagent (Invitrogen) according to the manufacturer's instructions. One microgram of total RNA was used for cDNA synthesis in a 20-μl reaction with an PrimeScript RT reagent Kit (#RR037A, Takara Bio). One microliter of the cDNA library was used in a 20-μl PCR. TB Green Premix (#RR820A, Takara Bio) was used to determine the threshold cycle value for each sample using the ABI Q7 Fast Real-Time PCR System according to the manufacturer's instructions. The β-actin mRNA (*Actb*) served as the normalization gene in these studies. ΔCt was calculated by the threshold cycle (Ct) for *Actb* minus the Ct for the target gene and the relative expression levels for the target genes determined by $2^{\Delta Ct}$ were visualized by GraphPad Prism 8.0.2. Primer sequences used for quantitative PCR are listed in Supplementary Table 2.

### Flow cytometry

Data was collected on a BD LSRFortessa flow cytometer using 488-1 (excitation at 488 nm, emission at 525/50 nm) for the sensor, and 561-2 (excitation at 561 nm, emission at 670/30 nm) for propidium iodide (PI) intensity. Twenty thousand fluorescent cells were evaluated per condition and cells were gated to exclude debris and doublets. The mean fluorescence intensity (MFI) was obtained by using the derived function on FlowJo software (FlowJo LLC).

### Adeno-associated virus (AAV) packaging

AAV was produced by using AAVpro Helper Free system (Takara Bio, #6230) according to the manufacturer's instructions with minor modifications. Briefly, DNA sequence encoding cBioITA or the control biosensor cdBioITA was subcloned into an AAV2 vector (Serotype 2, CD68/150 promoter). AAV particles were produced by co-transfecting AAVpro 293T cells with AAV2 plasmid containing coding sequence of itaconate biosensor, and two packaging plasmids pRC2-mi342 and pHelper. Infectious AAVs were extracted from AAV-producing 293T cells at day 3 post plasmid transfection and purified using an AAVpro purification kit (Takara Bio, #6232). AAV genome titer was determined by qPCR using an AAVpro titration kit (Takara Bio, #6233).

### AAV administration and endotoxin-induced sepsis model

Six-week-old female C57BL/6J mice were injected intravenously with 1 × 10¹¹ viral genomes of cBioITA/cdBioITA-expressing AAVs suspended in 100 μl PBS. At day 5 post AAV injection, LPS (1 mg kg⁻¹) suspended in 200 μl PBS or vehicle control (200 μl PBS) was delivered to mice via intraperitoneal administration. Mice were euthanized in a CO₂ chamber 8 h after LPS administration. For flow cytometric analysis of cBioITA/cdBioITA-expressing macrophages, mice peritoneal lavage was performed using 10 ml PBS, the cells were pelleted from the lavage fluid by centrifugation, resuspended in 1 ml PBS, and passed through a 70-μm filter. The cells were then stained with 1 μg ml⁻¹ of PE/Cyanine7 anti-mouse F4/80 and Pacific blue anti-mouse/human CD11b antibodies for 20 min at 4 °C. The samples were washed, resuspended in

200 μl PBS, and analyzed using a BD LSRFortessa flow cytometer. Acquired data were analyzed using the FlowJo software (FlowJo LLC). The cBioITA/cdBioITA-positive peritoneal macrophages are obtained by FACS-sorting using the same methods in flow cytometric analysis of cBioITA/cdBioITA-expressing macrophages, and then immunoblotting and LC−MS assays were performed to analyze the levels of IRG1 expression and intracellular itaconate in these cells.

All mice were housed under a specific pathogen-free conditions with the temperature maintained at $23 \pm 2\,°C$ and relative humidity at 50–65% under a 12 h/12 h light/dark cycle. The animals were housed at 3–5 mice per cage with free access to food and water. The use of animals in this study was approved by the Institutional Animal Care and Use Committee of the Institute of Biophysics, Chinese Academy of Sciences.

### Statistics and reproducibility

IBM SPSS statistics 23 and GraphPad Prism 8.0.2 were used to perform statistical analyses. All data are expressed as mean ± SD or mean ± SEM, as specified in figure legends. Sample numbers ($n$) and experimental repeats are indicated in figure legends. All immunoblotting results were presented as a representative example of a single experiment repeated at least in triplicate. For comparison of paired variables, group differences were analyzed using two-tailed Student's $t$-test. For comparison of multiple variables, group differences were analyzed using one-way ANOVA. Values of $P < 0.05$ were considered statistically significant.

### Reporting summary

Further information on research design is available in the Nature Research Reporting Summary linked to this article.

## Data availability

The atomic coordinates and structure factors generated in this study have been deposited in the Protein Data Bank under accession codes: "7W08" (overall structure of IBD in apo), "7W07" (IBD in complex with itaconate) and "7W06" (SeMet labeled IBD in complex with itaconate). The data supporting the findings of this study are available within the article and its Supplementary information file. Source data, including cell videos, numerical source data, structure validation reports, unprocessed gels and vector map files, are provided with this paper. Source Data are provided with this paper.

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

## Acknowledgements

We thank the staff of the BL17U1 and BL18U1 beamlines at the Shanghai Synchrotron Radiation Facility (SSRF) for their assistances during data collection. This work was supported by the Training Program of the Major Research Plan of the National Natural Science Foundation of China (Grant No. 92157104 to X.L.), the National Key R&D Program of China (Grant No. 2020YFC2002700 to X.L.), the Key Program of the Chinese Academy of Sciences (Grant No. KJZD-SW-L05 to X.L.), the National Natural Science Foundation of China (Grant No. 82073060 to X.L.), and the National Science Foundation for Young Scientists of China (Grant No. 82003032 to Z.Z. and Grant No. 82103349 to C.C.).

## Author contributions

This study was conceived by X.L. X.L., P.S., and Z.Z. designed the study. P.S., Z.Z., B.W., C.L., C.C., and P.L. performed the experiments. X.L., P.S., and Z.Z. wrote the manuscript with comments from all authors.

## Competing interests

The authors declare no competing interests.
