## [Peer Review File · Nature Communications]

REVIEWER COMMENTS

Reviewer #1 (Remarks to the Author):

In this manuscript, Sun et al. describe development and validation a genetically encoded fluorescent itaconate biosensor, BioITA, for directly monitoring itaconate dynamics in subcellular compartments of living macrophages. This biosensor has been applied to monitor the itaconate dynamics in response to lipopolysaccharide (LPS) stimulation. Also, it was used to show that STING activation induces itaconate production, and injection of viral particles expressing cytosolic BioITA into mice allows to observe elevation of itaconate level in activated macrophages derived from LPS-injected mice. Authors claim that the BioITA provides a broadly applicable sensitive tool for detecting itaconate with high spatiotemporal resolution in live cells and facilitates screening for drug or gene candidates that affect uptake, efflux, and metabolism of itaconate.

This is an interesting study that develops a new biosensor tool for detection of itaconate in macrophages. The biosensor development approach and its application could be of interest for the wider scientific community. However, there are several issues that need to be addressed before the manuscript could be considered for publication. These are listed below.

General comments:

The literature review is very limited with only 21 references cited. Background information on itaconate as important chemical and signalling molecule, macrophages and biosensors should be revised and discussed in more detail.

The information provided in the manuscript and methodology within could be improved by including more detailed information on how biosensor was built and how some experiments were performed. Since development of biosensor forms a major part of manuscript, provision of nucleotide sequences and vectors for constructing BioITA would be highly beneficial. The nucleotide sequences encoding itaconate-binding domain (IBD) (ItcR truncation of residues 80 to 292), circularly permuted green fluorescent protein (cpGFP), BioITA, and human IRG1 used for building BioITA should be provided and assembly strategies described. Manuscript also misses detailed description of mBioITA and cBioITA describing the tagging with organelle-specific signal peptides. Information of these two constructs should be clearly described.

Authors claim their data demonstrate that BioITA can reliably detect itaconate generation with high sensitivity in different cellular compartments of living macrophages. However, only cytoplasmic and mitochondrial compartments have been tested. Moreover, judging by Figure 3A and followed figures, reliability and resolution of itaconate monitoring in macrophages isn't high. Therefore, authors have to provide images with better resolution and additional data for other compartments or reconsider their statements.

Authors claims that data demonstrate that BioITA correctly reports the fluctuations of mitochondrial and cytosolic itaconate in LPS-activated. However, provided data are not sufficient to make such claim.

Finally, the claim that the BioITA provides a broadly applicable sensitive tool for detecting itaconate with high spatiotemporal resolution in live cells and facilitates screening for drug or gene candidates that affect uptake, efflux, and metabolism of itaconate should be revised or authors should provide additional data which demonstrates broad applicability of this tool, high spatiotemporal resolution and use in screening of drugs and gene candidates.

Specific comment:

Naming of supplementary figures needs to be standardized, as in the current version of submission, source data files are labelled with S index, e.g., Fig. S4, whereas these figures in the manuscript are referred without the S index, e.g. Extended Data Fig. 4.

References need to be revised making sure that the style is consistent.

Reviewer #2 (Remarks to the Author):

This is a very interesting and well-carried out study on a biosensor for detecting itaconate. I have two issues I would like to have addressed.

1. Can the sensor detect itaconate derivatives such as 4-OI and Diethyl-itaconate? These are widely used and so it would be interesting to know if the sensor can detect them. It would also be interesting to determine whether they give a signal if added to macrophages, which would also be interesting to determine in the case of adding itaconate itself to cells.

2. Recently 2 isomers of itaconate have been reported - mesaconate and citraconate. Can the sensor detect these too? This is important to determine, to ensure specificity.

Reviewer #3 (Remarks to the Author):

Manuscript by Sun et al describes design and testing of the genetically encoded itaconate sensor in relevant cells. Detection of itaconate in human/mammalian cells is clearly important and this sensor represents a valuable tool.

Authors have determined the 3D structure of the bacterial itaconate biosensor and detected a conformational shift, upon itaconate binding which is the basis for their biosensor. They have refined the design of a sensor by optimization of the length of the linker between the cpGFT and IBD achieving up to 3 fold at 4 uM ITA concentration, which seems a bit low but nevertheless sufficient for most experiments they performed. Response of the BioITA sensor convincingly responded to the appropriate stimuli (LPS, STING agonist) and KOs of the genes that affect ITA production or consumption. Results on mouse macrophages based on AAV based delivery of BioITA were done on

isolated macrophages which could have been done by direct infection of cells.

The manuscript is technically well performed, well written and relevant however I have identified the following issues that should be resolved:

1. Reversibility is mentioned but not really well demonstrated. In fact all experiment on cells demonstrate increasing amount of the signal without a single instance of its decrease. Authors mention that Fig2d (in vitro response of the sensor) shows reversibility but it is not clear if this was performed on the same sample or not. What is wash, since the sensor is a soluble protein ? Results should demonstrate activation of a sensor by ITA, decrease of the signal and again and increase by IITA, on the same sample. This is important since the reversibility of the fluorescent protein based sensors may be questionable due to the requirement for the maturation of the fluorophore.
2. Particularly on cells, since the stable lines were prepared reversibility should be easily demonstrated on a longer time scale after the signal for production of ITA has been removed.
3. Authors claim that the response of their BioITA sensor is fast, however in vitro (Fig 2e) it takes almost 1 hour to achieve the maximal amplitude, which seems slow for the system which should rapidly achieve an equilibrium. Could the authors comment on this. May it have to do with the maturation of the fluorophore ? Could substantially higher concentration of the ITA reach the equilibrium faster ?
4. Could the authors estimate the concentration of ITA in different cellular compartments based on the calibration curve ? Comparison of the cytosolic and mitochondrial sensor suggest that the concentration of ITA is higher in the cytosol than in mitochondria where it is generated. Even after blocking the transporter Ogc the response of the sensor (concentration of ITA) in the cytosol seems only slightly lower than in mitochondria.
5. It looks the sensor starts to produce the response to LPS stimulation almost instantly, which would suggest direct protein based activation rather than transcriptional regulation. Could the authors comment on this, as well as on the apparent delay of approx. 2 hours for the cytosolic sensor, which seems not to be present upon stimulation by 2,3-cGAMP.

Point-by-point response:

Reviewer #1 (Remarks to the Author):

In this manuscript, Sun et al. describe development and validation of a genetically encoded fluorescent itaconate biosensor, BioITA, for directly monitoring itaconate dynamics in subcellular compartments of living macrophages. This biosensor has been applied to monitor the itaconate dynamics in response to lipopolysaccharide (LPS) stimulation. Also, it was used to show that STING activation induces itaconate production, and injection of viral particles expressing cytosolic BioITA into mice allows to observe elevation of itaconate level in activated macrophages derived from LPS-injected mice. Authors claim that the BioITA provides a broadly applicable sensitive tool for detecting

itaconate with high spatiotemporal resolution in live cells and facilitates screening for drug or gene candidates that affect uptake, efflux, and metabolism of itaconate.

This is an interesting study that develops a new biosensor tool for detection of itaconate in macrophages. The biosensor development approach and its application could be of interest for the wider scientific community. However, there are several issues that need to be addressed before the manuscript could be considered for publication. These are listed below.

Answer: We greatly appreciate the reviewer's acknowledgement of the potential significance of this report and the insightful comments.

General comments:

The literature review is very limited with only 21 references cited. Background information on itaconate as important chemical and signalling molecule, macrophages and biosensors should be revised and discussed in more detail.

Answer: We have revised the manuscript following the reviewer's suggestions. More background information related to the immunomodulatory role of itaconate and detection of intracellular metabolite levels using biosensors has been added into the Introduction section of the revised manuscript. To track these modifications, the revised content was highlighted with yellow.

The information provided in the manuscript and methodology within could be improved by including more detailed information on how biosensor was built and how some experiments were performed. Since development of biosensor forms a major part of manuscript, provision of nucleotide sequences and vectors for constructing BioITA would be highly beneficial. The nucleotide sequences encoding itaconate-binding domain (IBD) (ItcR truncation of residues 80 to 292), circularly permuted green fluorescent protein (cpGFP), BioITA, and human IRG1 used for building BioITA should be provided and assembly strategies described. Manuscript also misses detailed description of mBioITA and cBioITA describing the tagging with organelle-specific signal peptides. Information of these two constructs should be clearly described.

Answer: We have revised the manuscript following the reviewer's suggestions. BioITA-expressing vectors and the related vectors have been listed in **Supplementary Table 3** and the vector's map files containing the plasmid backbone sequence and coding sequences of inserted genes are provided as source data. Assembly strategy for building BioITA and information for constructing mBioITA-, cBioITA-, and human/mouse IRG1-expressing vectors have been described in the Methods section with a subtitle "**Construction of expression vectors**" in the revised manuscript.

Authors claim their data demonstrate that BioITA can reliably detect itaconate generation with high sensitivity in different cellular compartments of living macrophages. However, only cytoplasmic and mitochondrial compartments have been tested. Moreover, judging by Figure 3A and followed figures, reliability and resolution of itaconate monitoring in macrophages isn't high. Therefore, authors have to provide

images with better resolution and additional data for other compartments or reconsider their statements.

Answer: The reviewer's point is well taken. To improve the accuracy of our statements, we have changed the original statement "...different cellular compartments..." to "...mitochondria and cytosol..." in the revised manuscript. In addition, images with better resolution have been provided in **Figures 3-5** of the revised manuscript.

Authors claims that data demonstrate that BioITA correctly reports the fluctuations of mitochondrial and cytosolic itaconate in LPS-activated. However, provided data are not sufficient to make such claim.

Answer: The reviewer's point is well taken. To correct this inappropriate statement, we have changed the statement "... BioITA correctly reports the fluctuations of mitochondrial and cytosolic itaconate ..." to "... BioITA correctly reports the increased levels of mitochondrial and cytosolic itaconate ..." in the revised manuscript.

Finally, the claim that the BioITA provides a broadly applicable sensitive tool for detecting itaconate with high spatiotemporal resolution in live cells and facilitates screening for drug or gene candidates that affect uptake, efflux, and metabolism of itaconate should be revised or authors should provide additional data which demonstrates broad applicability of this tool, high spatiotemporal resolution and use in screening of drugs and gene candidates.

Answer: The reviewer's point is well taken. To avoid the potential overstatement, we have changed the last sentence in the Abstract "Thus, BioITA provides a broadly applicable sensitive tool for detecting itaconate with high spatiotemporal resolution in live cells and facilitates screening for drug or gene candidates that affect uptake, efflux, and metabolism of this important anti-inflammatory metabolite." to "Thus, BioITA enables subcellular resolution imaging of itaconate in living macrophages." in the revised manuscript.

Specific comment:

Naming of supplementary figures needs to be standardized, as in the current version of submission, source data files are labelled with S index, e.g., Fig. S4, whereas these figures in the manuscript are referred without the S index, e.g. Extended Data Fig. 4.

Answer: We have corrected these mistakes in the revised manuscript.

References need to be revised making sure that the style is consistent.

Answer: Reference style has been corrected in the revised manuscript.

Reviewer #2 (Remarks to the Author):

This is a very interesting and well-carried out study on a biosensor for detecting itaconate. I have two issues I would like to have addressed.

Answer: We greatly appreciate the reviewer's positive comments.

1. Can the sensor detect itaconate derivatives such as 4-OI and Diethyl-itaconate? These are widely used and so it would be interesting to know if the sensor can detect them. It would also be interesting to determine whether they give a signal if added to macrophages, which would also be interesting to determine in the case of adding itaconate itself to cells.

Answer: To perform the reviewer's suggested experiments, we purchased itaconate derivatives 4-octyl itaconate (OI) and dimethyl-itaconate (DI) (PMID: 31178405), as well as itaconate isomers mesaconate and citraconate (PMID: 35655026; PMID: 35655024), but not diethyl-itaconate due to this compound is not locally available. Fluorescence emission scan assay (excitation at 488 nm) showed that OI and DI didn't alter the fluorescence from BioITA (Fig. 2f, g), suggesting that OI and DI, as itaconate derivatives, were not detected by BioITA. Intriguingly, cBioITA-expressing, but not the control biosensor cdBioITA-expressing, RAW264.7 cells exhibited marked increases of fluorescence upon treatment with unmodified itaconate (Supplementary Fig. 3g). In contrast, moderate increases of cBioITA's fluorescence were observed in RAW264.7 cells upon treatment with OI or DI (Supplementary Fig. 3g). These data imply that exogenous itaconate, as well as derivatized itaconate, is capable of elevating intracellular itaconate level of macrophages.

2. Recently 2 isomers of itaconate have been reported - mesaconate and citraconate. Can the sensor detect these too? This is important to determine, to ensure specificity.

Answer: We have performed the reviewer's suggested experiments. Mesaconate and citraconate were not detected by BioITA, as evidenced by fluorescence emission from BioITA in presence of mesaconate and citraconate (Fig. 2f, g). These data suggest that BioITA detects itaconate with a high specificity.

Reviewer #3 (Remarks to the Author):

Manuscript by Sun et al describes design and testing of the genetically encoded itaconate sensor in relevant cells. Detection of itaconate in human/mammalian cells is clearly important and this sensor represents a valuable tool.

Authors have determined the 3D structure of the bacterial itaconate biosensor and detected a conformational shift, upon itaconate binding which is the basis for their biosensor. They have refined the design of a sensor by optimization of the length of the linker between the cpGFT and IBD achieving up to 3 fold at 4 uM ITA concentration, which seems a bit low but nevertheless sufficient for most experiments they performed. Response of the BioITA sensor convincingly responded to the appropriate stimuli (LPS, STING agonist) and KOs of the genes that affect ITA production or consumption. Results on mouse macrophages based on AAV based delivery of BioITA were done on isolated macrophages which could have been done by direct infection of cells.

The manuscript is technically well performed, well written and relevant however I have identified the following issues that should be resolved:

Answer: We greatly appreciate the reviewer's acknowledgement of the potential significance of this report and the insightful comments, which are essential for the

improvement of this manuscript. The necessary experiments have been performed to address the reviewer's outstanding and constructive questions.

1. Reversibility is mentioned but not really well demonstrated. In fact all experiment on cells demonstrate increasing amount of the signal without a single instance of its decrease. Authors mention that Fig2d (in vitro response of the sensor) shows reversibility but it is not clear if this was performed on the same sample or not. What is wash, since the sensor is a soluble protein? Results should demonstrate activation of a sensor by ITA, decrease of the signal and again and increase by IITA, on the same sample. This is important since the reversibility of the fluorescent protein based sensors may be questionable due to the requirement for the maturation of the fluorophore.

Answer: The reviewer's point is well taken. In the original Fig. 2d, the reversibility of binding between itaconate (ITA) and BioITA was tested by the fluorescence scanning from the same BioITA sample before and after itaconate elution using phosphate buffered saline (PBS) in an Amicon Ultra tube with a 10 kiloDalton molecular weight cutoff filter. To further confirm the binding of itaconate to BioITA is reversible, we performed the reviewer's suggested experiments and obtained the results showing that itaconate exposure elevated fluorescence emission from BioITA and the repeated effect was observed following the itaconate elution from BioITA using an Amicon Ultra tube with a 10 kiloDalton molecular weight cutoff filter (Fig. 2d).

2. Particularly on cells, since the stable lines were prepared reversibility should be easily demonstrated on a longer time scale after the signal for production of ITA has been removed.

Answer: The reviewer's point is well taken. To provide evidence supporting that the binding of itaconate to BioITA is reversible in live cells, mouse IRG1 protein was exogenously expressed under the control of a tetracycline-inducible promoter in RAW264.7 cells stably expressing mitochondria- or cytosol-localized BioITA. Immunoblotting analysis indicated that pulse treatment with doxycycline for 30 min induced a transient expression of IRG1 during 4 to 8 hours post doxycycline treatment (Supplementary Fig. 3h). As expected, fluorescence intensity of mitochondrial or cytosolic BioITA in RAW264.7 cells exhibited a transient elevation during the time period of IRG1 expression (Supplementary Fig. 3i, j). In line with this finding, intracellular itaconate level exhibited a transient elevation during the time period of IRG1 expression, as evidenced by the biochemical assay (Supplementary Fig. 3k). Taken together, our data demonstrate that BioITA is able to detect fluctuations of mitochondrial and cytosolic itaconate in living macrophages, thereby supporting the binding of itaconate to BioITA is reversible in live cells.

3. Authors claim that the response of their BioITA sensor is fast, however in vitro (Fig 2e) it takes almost 1 hour to achieve the maximal amplitude, which seems slow for the system which should rapidly achieve an equilibrium. Could the authors comment of this. May it have to do with the maturation of the fluorophore? Could substantially higher concentration of the ITA reach the equilibrium faster?

Answer: The reviewer's point is well taken. To check the dynamic of IRG1-catalyzed itaconate production from *cis*-aconitate, the itaconate generation at different time points post active IRG1 addition was determined by liquid chromatography–mass spectrometry (LC–MS) analysis. In line with the results obtained from real-time detection using BioITA, itaconate level increased in a time-dependent manner and achieved a concentration (~ 2000 μ M) close to the maximal working concentration of BioITA at 60 min post active IRG1 addition (**Supplementary Fig. 2c**). These results suggest that BioITA may rapidly and reliably detect itaconate within its range of working concentration without significant limitation by fluorophore maturation.

4. Could the authors estimate the concentration of ITA in different cellular compartments based on the calibration curve? Comparison of the cytosolic and mitochondrial sensor suggest that the concentration of ITA is higher in the cytosol than in mitochondria where it is generated. Even after blocking the transporter Ogc the response of the sensor (concentration of ITA) in the cytosol seems only slightly lower than in mitochondria.

Answer: We have performed the reviewer's suggested experiments. To estimate concentrations of free intracellular itaconate, we accessed the fluorescence intensity of cytosolic BioITA in digitonin-permeabilized RAW264.7 cells using flow cytometry in the presence of external itaconate with determined concentrations. Digitonin-mediated cell permeabilization was assessed by influx assay of propidium iodide (PI) (**Supplementary Fig. 3d**). Based on the calibration curve generated by plotting the normalized mean fluorescence intensity (MFI) from cBioITA versus its corresponding equilibrated itaconate concentration (**Supplementary Fig. 3e**), we estimated that the concentration of free itaconate was 551 μ M [95% confidence interval (CI): 457 to 645 μ M] in mitochondria and 1757 μ M (95% CI: 1269 to 2245 μ M) in cytosol of non-permeabilized RAW264.7 cells stimulated with LPS for 12 hours (**Supplementary Fig. 3f**).

Given that itaconate generated in the mitochondrial matrix is exported to the cytosol, thus higher cytosolic itaconate (ITA) concentration may be caused by high-efficient efflux of itaconate from mitochondria to cytosol by itaconate transporters. In line with this hypothesis, the increase of mitochondrial itaconate concentration accompanied by the decrease of cytosolic itaconate concentration was observed when *OGC*, the gene encoding one of known itaconate transporter, was knocked out (**Fig. 3d, e**).

5. It looks the sensor starts to produce the response to LPS stimulation almost instantly, which would suggest direct protein based activation rather than transcriptional regulation. Could the authors comment on this, as well as on the apparent delay of app. 2 hours for the cytosolic sensor, which seems not to be present upon stimulation by 2,3-cGAMP.

Answer: The reviewer's point is well taken. Results from quantitative PCR (qPCR) assay showed that the level of *Irg1* mRNA was significantly upregulated at 2 hours post LPS stimulation (**Supplementary Fig. 5a**), supporting the notion that LPS stimulation

activates *Irg1* expression through transcriptional activation of this gene. In contrast, we observed that *Irg1* mRNA level exhibited a significant upregulation at 5 hours post 2,3-cGAMP stimulation (Supplementary Fig. 5a). These results demonstrate that 2'3'-cGAMP stimulation results in a delayed activation of *Irg1* expression compared to LPS stimulation, implying that 2,3-cGAMP is a less potent agonist for activation of *Irg1* expression compared to LPS.

Given that itaconate is generated in the mitochondrial matrix and then exported to the cytosol, thus the detection of cytosolic itaconate by BioITA may be observed a bit later than that of the mitochondrial itaconate. However, the delayed elevation of cytosolic itaconate is less obvious when the macrophages were stimulated with 2,3-cGAMP (Fig. 5a, b), that may be due to the less potency of 2,3-cGAMP for activation of *Irg1* expression.

REVIEWERS' COMMENTS

Reviewer #1 (Remarks to the Author):

I would like to thank the authors for addressing my comments and concerns. All major and minor points have been revised to the satisfactory level.

Reviewer #2 (Remarks to the Author):

The authors have adequately addressed my concerns and I'm happy to recommend acceptance.

Reviewer #3 (Remarks to the Author):

Authors have performed additional experiments that resolved the raised issues such as on the reversibility, including response to the pulse of induction and organelle concentration of the itaconate and clarified some issues. Therefore the manuscript has been improved and is my opinion suitable for publication.